# Polarization Conforms Performance Variability in Amorphous Electrodeposited Iridium Oxide pH Sensors: A Thorough Surface Chemistry Investigation

**DOI:** 10.3390/s24030962

**Published:** 2024-02-01

**Authors:** Paul Marsh, Mao-Hsiang Huang, Xing Xia, Ich Tran, Plamen Atanassov, Hung Cao

**Affiliations:** 1Department of Electrical Engineering and Computer Science, University of California Irvine, Irvine, CA 92697, USA; marshp@uci.edu (P.M.); maohsiah@uci.edu (M.-H.H.);; 2Irvine Materials Research Institute, University of California Irvine, Irvine, CA 92697, USA; ictran@uci.edu; 3Department of Chemical and Biomolecular Engineering, University of California Irvine, Irvine, CA 92697, USA; plamen.atanassov@uci.edu; 4Department of Materials Science and Engineering, University of California Irvine, Irvine, CA 92697, USA; 5Department of Biomedical Engineering, University of California Irvine, Irvine, CA 92697, USA; 6Department of Computer Science, University of California Irvine, Irvine, CA 92697, USA

**Keywords:** iridium oxide, pH sensors, XPS, surface chemistry, variability

## Abstract

Electrodeposited amorphous hydrated iridium oxide (IrOx) is a promising material for pH sensing due to its high sensitivity and the ease of fabrication. However, durability and variability continue to restrict the sensor’s effectiveness. Variation in probe films can be seen in both performance and fabrication, but it has been found that performance variation can be controlled with potentiostatic conditioning (PC). To make proper use of this technique, the morphological and chemical changes affecting the conditioning process must be understood. Here, a thorough study of this material, after undergoing PC in a pH-sensing-relevant potential regime, was conducted by voltammetry, scanning electron microscopy (SEM), energy-dispersive X-ray spectroscopy (EDS), X-ray diffraction (XRD), and X-ray photoelectron spectroscopy (XPS). Fitting of XPS data was performed, guided by raw trends in survey scans, core orbitals, and valence spectra, both XPS and UPS. The findings indicate that the PC process can repeatably control and conform performance and surface bonding to desired calibrations and distributions, respectively; PC was able to reduce sensitivity and offset ranges to as low as ±0.7 mV/pH and ±0.008 V, respectively, and repeat bonding distributions over ~2 months of sample preparation. Both Ir/O atomic ratios (shifting from 4:1 to over 4.5:1) and fitted components assigned hydroxide or oxide states based on the literature (low-voltage spectra being almost entirely with suggested hydroxide components, and high-voltage spectra almost entirely with suggested oxide components) trend across the polarization range. Self-consistent valence, core orbital, and survey quantitative trends point to a likely mechanism of ligand conversion from hydroxide to oxide, suggesting that the conditioning process enforces specific state mixtures that include both theoretical Ir(III) and Ir(IV) species, and raising the conditioning potential alters the surface species from an assumed mixture of Ir species to more oxidized Ir species.

## 1. Introduction

The uses of pH electrodes in the literature feature a host of transduction methods, interface technologies, and materials [1], although iridium oxide (IrOx) has been intensively studied as an electrolyte-contact electrochemical pH transducer for many years [2]. Out-of-cleanroom processes (i.e., electrodeposition, sol–gel deposition, and thermal oxidation, to name just a few) have been of interest given their inherent simplicity, low energy use, and lower cost as compared with thermal, sputtering, and similar methods. An additional factor in favor of electrodeposition is that many of these protocols produce amorphous hydrated IrOx films, which are generally ‘super-Nernstian’ (i.e., bulk materials featuring sensitivity slopes above the Nernstian scaled thermal voltage relationship of 59.1 mV/pH). Our group and others have demonstrated many applications and complete systems [3,4,5,6,7,8,9]; however, the performance of electrodeposited IrOx pH sensors is still inconsistent for rigorous research use and commercialization [6].

IrOx electrodes show large variances in initial sensitivity and comparatively rapid degradation, both of which are poorly understood. Even speciation [10,11] and deposition processes [12] are still under debate. Conventional pH sensors, such as a glass-encased dual-junction Ag/AgCl, feature relatively constant sensitivity due to saturated electrolyte concentrations at the reference electrode (RE) and membrane-restricted ionic passage at the working electrode (WE). In contrast, IrOx WE transducers face pronounced standard potential and sensitivity drift due to non-reversible reactions taking place on the material’s surface, more than slightly less sensitive devices [13,14]. Additionally, they feature performance variation within batches of as much as 10% [5]. Electrodeposition features variation regardless of environmental controls, digitized deposition programs, and the batch fabrication of substrates. Fabrication and performance variability are likely correlated, given an assumption that the surface state mixture is a result of final deposition potentials. Understandings of both variation and degradation are generally suggested to result from a variety of mechanisms, such as the oxidation state of Ir ligands [15,16] and specific terminating groups [17], the presence and/or diffusion of oxygen [18], and the dissolution of surface ligands [19,20,21], among others. It has been theorized by several authors that the ‘super-Nernstian’ sensitivity of the material stems from non-unity ratios of the proton to electron released from different oxidation states during ion adsorption [15,22]; in particular, IrOx’s ‘super-Nernstian’ sensitivity has been suggested to be correlated specifically to Ir(III)/Ir(IV) mixture quantities [23]. These oxidation states are likely a direct result of the electrode potential inducing oxidation reactions of IrOx surface ligands [17], thus altering reaction stoichiometry. Taken together, the surface oxidation state (i.e., terminating species) should be thought of as linked to pH sensitivity via stoichiometry, and the alteration of this state as contributing to degradation.

This variability calls for an in situ rehabilitation process to stabilize and enhance performance. Therefore, surface speciation control is ideal. It has been known for some time that a degree of control over pH sensitivity can be exercised by holding IrOx probes at specific potentials in a conductive medium, known as a preconditioning stage [16]. This will be referred to hereafter simply as ‘potentiostatic conditioning’ (PC), as it needs not be performed exclusively prior to transduction. This technique has been demonstrated on amorphous electrodeposited IrOx probes [24], likely viable across the material’s use lifetime [6], assuming that consistent electrolytes are used. Carroll et al. used potentiostatic operations to control for variations in both standard potential and sensitivity, both before and during use. It can be argued that this demonstrated self-calibration, as the slope and offset of transducer calibration was enforced to a window within their specific fabrication techniques and morphologies.

As performing PC increases the potential for ligand dissolution, clear understanding is imperative. To that end, the focus of this study was to understand and interpret the correlation between material state and electrode conditioning potentials within pH sensing regimes. Specifically, several compositional and morphological analyses were conducted on electrodeposited IrOx pH sensors to understand chemical variation. X-ray diffraction (XRD), scanning electron microscopy (SEM), and electron-dispersive X-ray spectroscopy (EDS) data were included to investigate whether the PC process affected morphology or uniformity across the sample surface. Surface chemistry investigations were performed via X-ray photoelectron spectroscopy (XPS) to determine how bonding and constituents were affected by applied conditioning voltages (Eappl), in relevant pH regimes. A quantitative analysis of these trends is supplied; a comprehensive discussion of the experimental procedure is followed by a thorough and self-consistent fitting procedure seeking to assign state mixture trends to correlated polarization potentials. General assignment of speciation was derived from literature, and quantification was derived from the fitting results. The paper is concluded with a discussion of suggested mechanisms. These data will show that the PC process maintains and conforms probe performance by repeatably altering surface bonding. In addition, based on common literature assumptions, the data are suggestive of amorphous IrOx thin films being repeatably oxidized to higher Ir oxidation states as conditioning potentials within this window increase.

## 2. Materials and Methods

### 2.1. Substrate Fabrication

IrOx films were fabricated on SiO_2_/Si wafer substrates with patterned metal layers in this study. The Si wafers were 100 mm in diameter, 500 μm thick, single-side polished, with a (100) orientation, and had a 500 nm layer of thermally grown SiO_2_ (University Wafer, Inc., Boston, MA, USA). Before use, these wafers underwent a solvent-cleaning (acetone, IPA, and N_2_ drying) process. The patterning process utilized a conventional lithography method. The wafers were coated with NR9-1500PY (Futurrex, Inc., Franklin, NJ, USA) photoresist and underwent a series of processes: baking, exposure to UV light using a mask, and development. Metal layers were added by sputtering or electron beam evaporation, depositing layers of chromium and gold. After metal deposition, the photoresist was stripped away, and additional layers were applied for insulation (Parylene-C) and as an etching mask (S1827). Parylene-C was vapor-deposited (PDS 2010, SCS, Indianapolis, IN, USA), followed by applying S1800 photoresist similarly to the initial layer. Some blistering under the exposed S1827 was noted, but it seldom impacted the process. The Parylene-C layer was etched using reactive ion etching. The wafers were sectioned into individual probes in the final stages, and any residual S1800 was cleansed away. Eliminating all polymer residues above the gold layers was critical for successful electrodeposition. The completed sample pads were 1 mm × 1 mm in size.

### 2.2. Electrodeposition Process

Electrodeposition of IrOx films was carried out using cyclic voltammetry, where probe test pads were immersed in an iridium oxalate solution. This method was preferred over pulsed deposition because it produces uniform and well-adhered films [25]. The iridium oxalate solution, easy to make and store, was prepared following Yamanaka’s method [26]: IrCl_4_ (ArtCraft Chemicals, Altamont, NY, USA) was mixed with water, followed by H_2_O_2_ (H325, Fisher Chemical, Vernon Hills, IL, USA) and oxalic acid dihydrate (247537, Sigma Aldrich, Burlington, MA, USA), and finally, the pH was adjusted with K_2_CO_3_ (P1472, Sigma Aldrich). This solution could be stored in a refrigerator for several months. Au/SiO_2_ probes served as working electrodes for the deposition process, with Ag/AgCl electrodes as a reference and platinum wire as counter electrodes. The deposition settings on a commercial potentiostat involved voltage sweeps between −0.5 V and +0.7 V for 1000 cycles at a rate of 5 V/s, taking about 8 min per coating. The potentiostat was pre-checked and calibrated before each use. All glassware used for mixing and testing was thoroughly cleaned with a nitric acid and hydrogen peroxide mixture, followed by boiling water baths and nitrogen drying. Glassware was stored carefully to ensure cleanliness.

### 2.3. Methods

Methods details are provided in the Appendix A, although a few notes are worth repeating here. First, sample substrates were lithographically patterned (see Appendix A) and electrodeposition was performed in batches using a potentiostat. All analysis performed in this study (voltammetry, SEM, EDS, and XRD) was conducted on this controlled pad area. Deposition was performed by cyclic voltammetry and the subsequent PC process was conducted in 10× phosphate-buffered saline (polarized at a chosen voltage for 180 s; Appendix A). Samples were then kept under vacuum and away from light until they could be mounted and transferred into XPS (~18–36 h for comparison samples) and analyzed when sufficient vacuum was reached. Finally, adventitious carbon (AdC) charge referencing was performed (aligning the main C 1s peak to an arbitrary position of 284.8 eV; pre-referenced C 1s spectra in Appendix A) for some spectra shown here; explicit text is provided where this technique was used, and no conclusions were then drawn from absolute binding energies.

### 2.4. XPS Experimental Flow

XPS was chosen for two reasons: the importance of surface chemistry to this transduction technique, and a lack of surface chemistry polarization trend data; most of the literature focuses on crystalline IrO_2_, or bulk chemistry. Ar+ etching was not used for contaminant removal due to surface homogeneity and reduction sensitivity (Appendix A), along with some preferential sputtering. Given the ongoing academic discussions of some spectra presented herein, O 1s detail spectra were taken prior to Ir 4f to minimize processing alterations.

Three sets of X-ray photoelectron spectroscopy (XPS) data were collected to account for surface variations possibly caused by light and oxygen exposure [27,28,29]. The first set involved probes coated and polarized at two different potentials per day, with measurements taken over three sessions. The second set had all probes coated and polarized in one day and scanned in a single session. The third set, focusing on a narrower voltage range, had probes coated and polarized in one day and scanned in one session. Each set used different potential increments: the first two sets ranged from −0.2 V to +0.8 V in 200 mV steps, while the third set ranged from +0.3 V to +0.8 V in 100 mV steps. Conditioning was applied to some probes before the XPS measurements. The first experiment established baseline measurements for the probes. Notable alterations in the XPS data were observed starting at +0.4 V. Care was taken to minimize contamination and air exposure during sample preparation. Samples were kept under a low vacuum and away from light until XPS analysis. The time from sample preparation to XPS measurement was controlled, and samples were exposed to air for no more than 30 min before being loaded into the XPS system. An additional dataset was used to determine the iridium-to-oxygen atomic ratios. These samples were prepared and analyzed over about two months, but were not made in consecutive deposition and polarization operations. This dataset is mentioned explicitly in the study.

Further discussion, significant additional fabrication and experimental details are provided in the Appendix A.

## 3. Results and Discussion

### 3.1. PC Performance Control

Voltammograms produced during consecutive depositions feature notable variation, when fabricated by either pulsed potential deposition (Appendix A) or cyclic voltammetry (Appendix A); all probes in the main paper were produced by cyclic voltammetry for uniformity and adhesion [25,26]. The cyclic voltammograms of Appendix A were produced consecutively in identical conditions, although still featured final peak current variations in approximately 40%, peak potential shifts of 40–70 mV, and redox couple reversibility reductions by up to 130 mV.

Figure 1 showcases PC outcomes from probe series (*n* = 3) polarized every 200 mV from −0.2 V to +0.8 V, vs. Ag/AgCl (in 1 M NaCl). Figure 1a demonstrates a selection of series across the potential range immediately after deposition; initial calibrations are within ±3.1 mV/pH and offset of ±60 mV. Upon conditioning, spreads drop to as little ±0.7 mV/pH and offset of ±8 mV, depending on PC potential (Figure 1b). As Figure 1c demonstrates, the linear regression error (i.e., buffer reactivity) is minimized with PC at +0.2 V. Figure 1d (averages of *n* = 3) shows drops in the offset range and standard deviation for every PC voltage chosen, indicating improvements in calibration clustering regardless of potential. Commensurate with the assumption that buffer reactivity is being altered, voltage drift upon immersion (Figure 1e) is also affected by the chosen PC potential. See the Appendix A for buffer constituents.

### 3.2. Morphology and Elemental Composition

Appendix A demonstrates the SEM, EDS, and XRD analyses of polarized films. GI-XRD stack plots in Appendix A indicate only minor Miller index differences between unpolarized and polarized films. While Appendix A includes reference lines for IrO_2_ Miller indices [30], only low-amplitude and high-width amorphous phase peaks are apparent, the most suggestive of which are at 50 degrees. This phase appears consistently regardless of PC potential, and does not appear to coincide with known Ir or IrO_2_ peaks. SEM and optical imaging (Appendix A) show areas of high electron return contrast, referred to hereafter as ‘shapes’; these shapes are consistently ~100 nm, independent of the polarization potential, and feature similar clustering. Regional areal density differed randomly, so comparison between potentials was not viable. Morphological trends held for unpolarized vs. polarized samples (Appendix A), suggesting that these are a feature of the deposition process. Given XRD’s disperse nature, crystallinity at the meso-/macroscale is not definitively rejected, but is also not supported. EDS maps in the same figures display no notable localization of Ir density following polarization. Bulk elemental data (Appendix A) are consistent across voltages and only expected elements are present: film (Ir and O), conductive substrate (Au and Cr), insulating substrate (Si and O), and some processing contamination (C). Additionally, Ir and Au relative peak intensities (insets) do not trend with polarization potential and are consistent with each other (see Appendix A for penetration depth justification). Taken together, XRD/SEM/EDS data suggest that the PC process makes only minor, and highly consistent, morphological changes to deposited films. In addition, there is no discernible variation in meso-/macro-scale crystallization, bulk elemental makeup, or nanoscale features in the thin films as polarization potential varies. Given the variation control featured in Figure 1, bulk morphological order is clearly not the predominant factor correlated to performance.

### 3.3. Surface Bonding Investigation

#### 3.3.1. Raw Polarization Trends

Figure 2a–d show O 1s and Ir 4f orbitals measured at points up to 42 days after fabrication (AdC charge referencing was performed for this figure; see method text and the Appendix A for discussion). O 1s peak intensity features a binding energy (B.E.) shift down with increasing time, while the Ir 4f peak intensity is stable. However, both orbitals experience shifts in intensity distribution: O 1s tends to gain intensity at lower binding energies, while Ir 4f gains intensity at higher binding energies. This group and others [6,24] have shown that aging in various conditions is correlated to performance variation; given PC’s variation control, the PC process likely affects surface chemistry. Figure 3 presents XPS trends across a range of polarization potentials, as compared with a freshly fabricated but unpolarized probe. Survey spectra in Figure 3a differ mainly in Na 1s at ~1070 eV (Appendix A), demonstrating consistent core orbital returns across potentials. As the potential increases, O 1s (Figure 3b) transitions shift intensity distribution and peak intensities lower in B.E., while Ir 4f doublets (Figure 3c) tend to shift their intensity distribution higher. Ir 5p transitions (Figure 3d) see only minor shifts to higher B.E.; this stability suggests that actual shifts in Ir 4f’s B.E. are minimal. These data suggest that the choice of PC potential can have a large effect on the resultant surface bonding. Core and valence spectra comparisons between unpolarized and high-potential polarized films (Figure 3 insets) show major alterations, despite the main figure deviations between unpolarized and low-potential polarized films being minimal. This suggests that “safe” potentials (i.e., minimal altering of surface bonding) are between 0.0 V and +0.4 V.

Many authors agree that the primary redox couple present in electrodeposition voltammograms (Appendix A) represents an Ir(III)/Ir(IV) pair [12,23,31] (there is little evidence for the existence of stable Ir(V) species); hence, it is assumed that both Ir(III) and Ir(IV) are present in thin films. Kötz et al. presented a cycle model [31] to explain how charge storage and oxygen production work in activated iridium oxide film (AIROF) electrodes, using insights from their XPS research. The model suggests that oxygen evolution is linked to a series of deprotonation steps (removal of hydrogen ions) during the chemical change of iridium hydroxide (Ir(OH)_3_) to a highly oxidized form of iridium (Ir(VI)) in iridium trioxide (IrO_3_). Additionally, the absorption of a water molecule can reverse this oxidation, turning the Ir(VI) back to a four-valent (tetravalent) state, thus completing the cycle. This cycle model offers a potential explanation for the processes occurring in these electrodes. Although there is some disagreement on whether an oxidized Ir will shift higher or lower in B.E. [32], the literature cited herein generally agrees that a more oxidized Ir manifests at a higher B.E. Given that ligands are generally assumed to be oxygen-based in this material, and O 1s and Ir 4f tend toward each other in B.E. with increasing PC potential, it is assumed that Ir has converted some portion of its formerly Ir(III) species to Ir(IV) via bonding changes with its oxygen ligands, or changes in the ligand terminations.

The XPS valence trends in Figure 3e feature bandgaps, which decrease, and secondary features (~7 eV), which gain amplitude relative to the primary (2.4 eV) feature at higher PC potentials. These indicate not only the possibility of satellites, but also suggest that they decline in prominence as increased polarization makes the surface more metallic. Relative valence feature amplitudes imply increased hybridization at higher potentials; the projected densities-of-state (PDOS) literature proposes the secondary valence feature to possibly represent the hybridization of Ir 5d and O 2p [33], so the valence feature nearest EF should represent free Ir 5d electrons [34,35]. The trends then indicate a conversion from free Ir valence electrons to more hybridized valence bonds with increased PC potential. Altogether, increased PC potential induces an alteration of species ratios to generally more oxidized Ir (increased hybridization of Ir–O bonding).

Interference with these trends is unlikely. First, Ir 5p’s overlap with 4f is invariant; if modeled by a 3/2, 1/2 spin orbit split doublet [36], 5p 1/2 overlaps with 4f detail spectra, but 3/2 is highly stable and therefore indicates that 1/2 is as well. Second, surface contaminants from PC (such as the Na Auger–Meitner line at ~1070 eV; Figure 3a and Appendix A) only minimally overlap with transitions of interest (such as O 1s) [36] and are uniform across voltages. Therefore, Ir 4f’s changes are predominantly correlated to its own intensity distribution.

One caveat is that crystallization is expected. Given that high polarization potentials are assumed to produce oxidized forms such as IrO_2_, and these more oxidized forms are more crystalline than their hydroxylated counterparts [10,28], crystallinity should increase with the polarization potential. As discussed above, GI-XRD on these films is unable to detect crystalline IrOx phases; additional future characterization is required. However, both core orbital and valence XPS trends are still indicative of useful guidelines for sensor variation control.

Figure 4 presents spectra of three probes prepared weeks apart and receiving PC conducted at the same potential. AdC charge referencing was performed to illustrate distribution patterns, so only relative binding energy shifts should be considered. Although there is minor widening of the Sample 2 spectra, intensity distribution shifts are minimal and repeatable. The conclusion can be drawn that the PC process, within the aforementioned potential window, is likely to induce consistent speciation (mixture of oxidation states) changes in amorphous electrodeposited IrOx in a variety of ages and surface bonding conditions, thus being a reliable means of controlling for performance variation; in other words, the PC process at an appropriate potential can consistently enforce distributions similar to recently fabricated films.

#### 3.3.2. Fitting and Quantification

For brevity, the complete XPS fitting procedure is presented in the Appendix A. However, aspects of this procedure are highly important to quantitation justification, and so are summarized here. IrOx’s varying semiconductor/metallic behavior [37,38] precluded work function charge referencing [39]; spectra are presented as-measured, or AdC-referenced where explicitly noted. Raw structure (Appendix A) guided the procedure, emphasizing simultaneous fitting and trend matching between potentials, and between core orbitals believed to be bonded. O 1s components (assumed hydroxide/water/Na 1s Auger–Meitner [31,40,41,42,43]) were simultaneously fit and locked to locate a potential oxide component (Figure 5a,c,e and Appendix A); patterns and components were then transferred in a similar procedure to Ir 4f, including satellite doublets (Appendix A) with minor LF asymmetry (Figure 5b,d,f and Appendix A). One assumption of note is that the Ir 4f “oxide” component B.E. shift direction was chosen to be higher with increased potential [10,11,17,21,44,45,46]. Another was that, while satellite trends could be estimated from valence analysis (Figure 3e, Figure 6 and Appendix A), starting values could not be. Figure 6 contains rough valence spectra fits and quantitation, which indicate a trend away from near-EF features with increased potential, but exact B.E. cannot be extracted from valence spectra. However, EELS plasmon peaks (which can be correlated to satellite offset [47,48,49]) for IrO_2_ [50], and PDOS simulations [10,11,46,51], yielded 1–2 eV seed values (Appendix A). Ir/O ratios (discussed below) and O 1s/Ir 4f species ratios support satellite assignment and are consistent with each other. All assumptions and justifications are discussed in detail in the Appendix A.

Final fits are shown in Appendix A, using fit parameters from Appendix A. Components assigned to O 1s as water bonds and overlapping Na Auger–Meitner lines are essentially static, while hydroxide and oxide ligand bond states transition from majority hydroxide to majority oxide with increasing potential. Ir 4f quantities provide quantitative self-consistency, in that hydroxide and oxide trends also reverse and are of similar percentages (doublets summed). Ir/O ratios retrieved from survey spectra (Appendix A) indicate an increasing proportion of O to Ir with increasing potential, from under 4:1 to over 4.5:1, supportive of the notion of higher quantities of hybridized Ir valence bonds over free Ir valence electrons. The increased hybridization of Ir–O bonding and increased O fits well with this assumed redox state change, given the assumption of Ir(III) correlating to hydroxy species and Ir(IV) correlating to oxide species [31,52]. However, as area ratios of component fits drawn from core level transitions are quantitatively different from atomic Ir/O ratios drawn from survey spectra, ligand speciation was considered to be significantly more complicated than simple O addition or removal, and so no specific molecular structure is indicated here.

In addition to the Ir 4f shift assumptions and XRD caveats enumerated above, mechanistic and speciation uncertainties remain for several reasons. First, some studies concerning metallic Ir films or solid metallic Ir which have been oxidized [31] feature unchanging Ir/O ratios across polarizations, indicating that perhaps the mechanism involved is not constant among all morphologies of IrOx. Second, work–function spectra are only mildly suggestive and have multiple interpretations. Differences between initial and final polarization (although non-linear) in the He I cutoff suggest a general decrease in work function (Appendix A); given that the secondary voltammogram peak pair in Appendix A is assumed by some to be an O^2−^/O^−^ redox couple [10,12], it is possible that work function decreases coincide with an increase in reduced terminating O, an increase in hydroxylated O^1−^ at the expense of terminating O, or simply an increase in Ir(IV) at the expense of Ir(III). Third, hydroxide/oxide ratios within O 1s and Ir 4f match each other in trend but differ in scale. Applications of area ratios between the species assumed to hydroxides and oxides, from O 1s to Ir 4f, produced a low residual initial fit, and both O 1s and Ir 4f fitted well when species assumed to be hydroxides gave way to species assigned to be oxides; however, lower polarization fits differed in ratio by an order of magnitude. Fourth, Ir/O ratios were too variable to denote an exact molecular makeup. Finally, residuals suggested a final component at 67.7 eV but, given that component assignment was linked here to oxidation state, and that Ir(V) is not assumed to be stable, residual area had to be covered by LF parameter extension. While fit stability is high, the data presented here are not quantitatively conclusive on speciation.

## 4. Conclusions

Despite continued uncertainty in exact speciation due to the limitations of current analytical techniques and the inherent complexity of the systems under study, the results here are valuable. They lend significant credibility to theories in the literature that the surfaces of amorphous electrodeposited IrOx materials are composed of a mixture of hydroxide and oxide ligand states, given the self-consistent quality of fits and their quantitative outputs, and that the conditioning process enforces surface oxidation states [17,21,44]. They demonstrate quantitative trends, from both an atomic ratio and a bond distribution standpoint, that the surface ligands are oxidizing at increased polarization potentials. They also suggest that the hybridization of orbitals from the two transitions of interest (Ir and O) occurs at increased rates with increased polarization. The study is valuable from an engineering perspective, providing important clues about the effectiveness of, and parameter choice for, performance variability control in amorphous IrOx pH sensors; it presents the PC process as a viable method of long-term variation control via surface speciation enforcement. While voltages below 0.0 V and above +0.4 V fall within contact pH sensing regimes, they do not appear to be suitable for self-calibration conditioning potentials; more extreme potentials transition the material away from surface state mixtures theorized to lead to “super-Nernstian” behavior (i.e., the initial mixtures after fabrication) and towards a singular composition. From an applied science perspective, the general trends shown are elucidative for surface chemistry in the potential sweep up towards the oxygen evolution reaction (OER). This makes them of interest to energy storage, fuel cell, and electrolysis catalyst research, for which amorphous IrOx is already academically interesting [53,54,55,56]. The spectra herein provide baseline data for future ion deposition studies in controlled environments to further explore surface reaction mechanisms. With a strictly confined deposition environment and substrate surface morphology, the proposed method can lead to the fabrication of hydrous IrOx pH sensors in a more controllable and predictable process, hence improving the repeatability and robustness of the sensor.

## Figures and Tables

**Figure 1 sensors-24-00962-f001:**
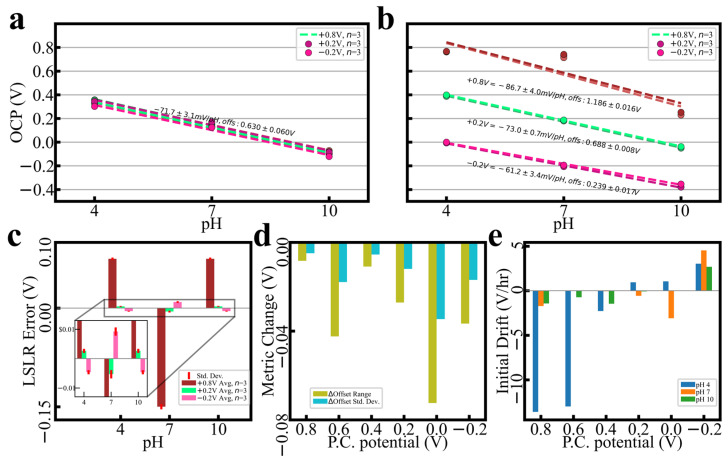
Demonstration of PC outcomes from a variety of potentials. (**a**) Calibration groupings prior to PC (legend enumerates PC potentials to align with the following subfigure); least-squares linear regressions (LSLR) and Pearson coefficients are supplied. (**b**) Calibration groupings following the PC process at selected potentials, along with recalculated LSLR and Pearson coefficients. (**c**) Errors as measured against post-PC LSLR; inset provided to show low-error detail. Standard deviations across *n* = 3 series provided. (**d**) Change in LSLR offset range and standard deviation after PC; negative values indicate decreases in each metric (i.e., an improvement in the metric by way of reduced variation). (**e**) Initial potential drift upon buffer immersion after PC, as measured in 10 s intervals.

**Figure 2 sensors-24-00962-f002:**
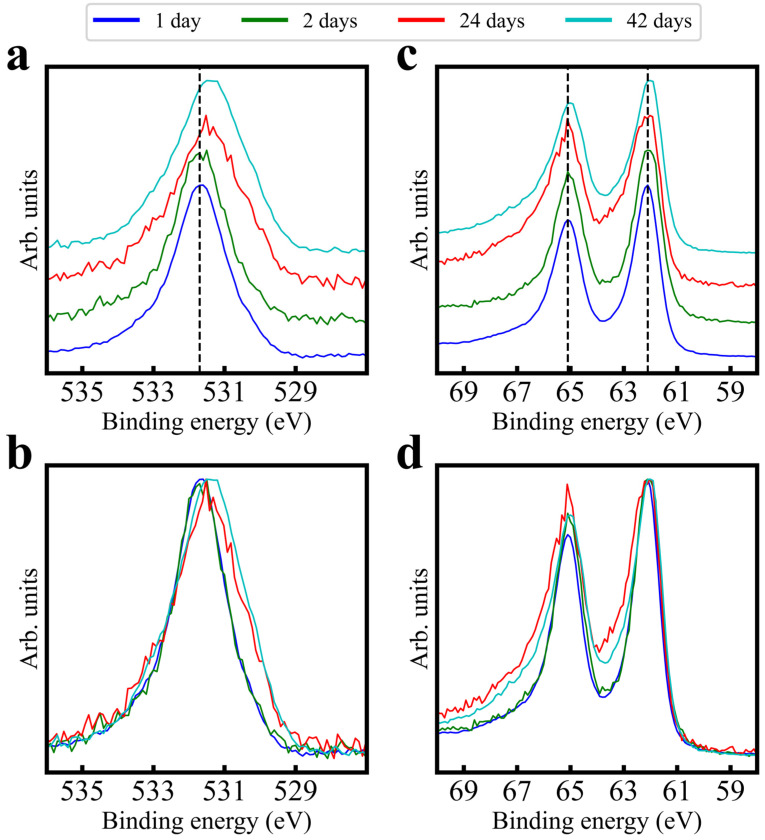
Stacked and overlaid XPS data, demonstrating binding energy shifts and intensity redistribution in the O 1s (**a**,**b**) and Ir 4f (**c**,**d**) spectra of samples measured a few days after fabrication, and samples fabricated 24 and 42 days after fabrication. All spectra were unity-based normalized. The data presented here have been arbitrarily AdC charge-referenced to 284.8 eV; thus, only relative binding energy shifts should be considered.

**Figure 3 sensors-24-00962-f003:**
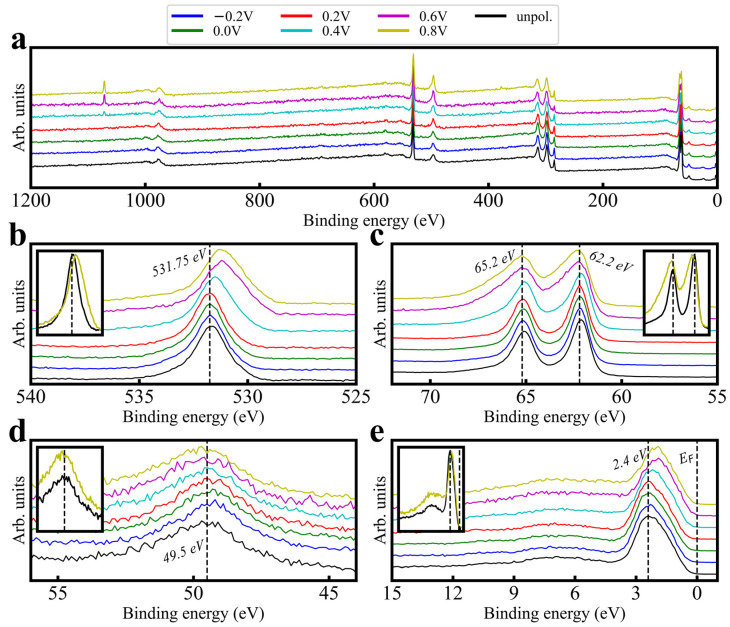
XPS spectra taken from a wide voltage range sample set fabricated consecutively and measured within a 24 h period. Spectra included are (**a**) survey spectra, (**b**) O 1s, (**c**) Ir 4f, (**d**) Ir 5p spectra, and (**e**) XPS valence (1486.6 eV). All spectra have been unity-based normalized and stacked. Insets are overlaid rather than stacked.

**Figure 4 sensors-24-00962-f004:**
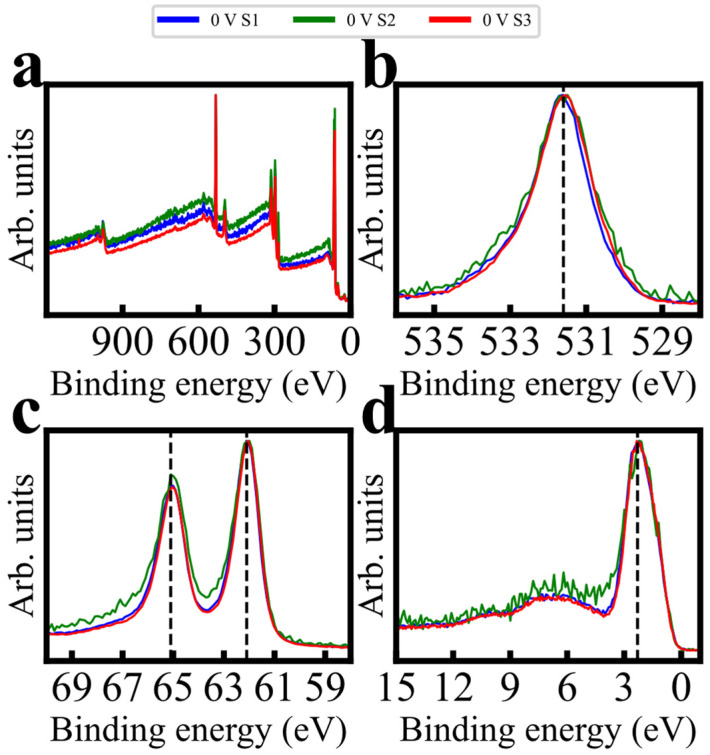
Consistency in XPS spectra across three samples polarized at 0 V, deposited, and polarized across a ~2 month span (from three separate fabrication batches). Spectra included are (**a**) survey, (**b**) O 1s detail, (**c**) Ir 4f detail, and (**d**) XPS excitation valence spectra. All spectra were unity-based normalized. The data have been arbitrarily AdC charge-referenced to 284.8 eV, so only relative binding energy shifts should be considered.

**Figure 5 sensors-24-00962-f005:**
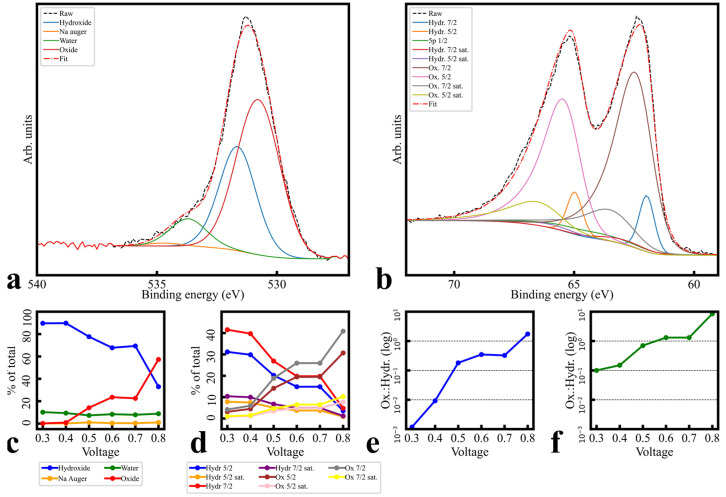
Fitted spectra of (**a**) O 1s and (**b**) Ir 4f, +0.8 V polarization of the single-day, narrow-voltage dataset, using the parameters listed in Appendix A and the procedure discussed in the main text. Satellites are included in (**b**). (**c**,**d**) Relative area ratios of components included in these fits for O 1s and Ir 4f fits, respectively. Satellites are included in (**d**). (**e**,**f**) Oxide/hydroxide cumulative component area ratios for O 1s and Ir 4f fits, respectively.

**Figure 6 sensors-24-00962-f006:**
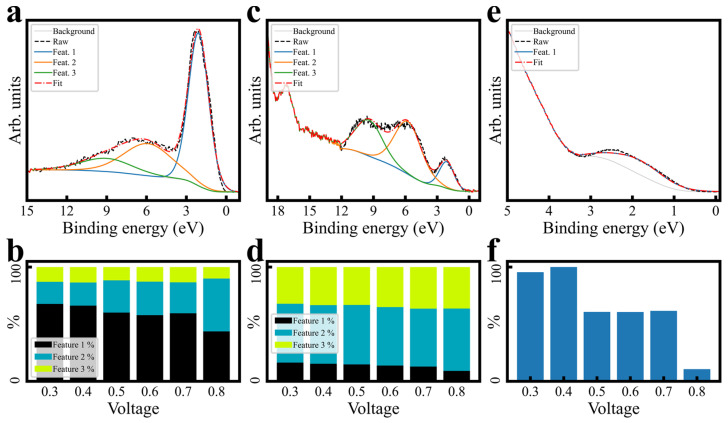
Rough quantitative analysis of valence spectra. Example fits (at +0.6 V polarization) are shown for XPS valence (**a**), UPS He II valence (**c**), and UPS He I valence (**e**). Relative feature ratios are pictured for XPS valence (**b**) and He II valence (**d**). Since He I valence was roughly fit with only a single component, (**f**) represents the area ratio for each fit compared with the maximum sized component for the whole series (<+0.4 V polarization, in this case). Valence fits across polarizations are presented in Appendix A.

## Data Availability

Data are contained within the article or Appendix A.

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
