# Peer review of "Polarization Conforms Performance Variability in Amorphous Electrodeposited Iridium Oxide pH Sensors: A Thorough Surface Chemistry Investigation"

_sensors, 2024, doi:10.3390/s24030962_

Round 1

Reviewer 1 Report

Comments and Suggestions for Authors

The authors describe a thorough study of the material IrOx, which underwent potentiostatic conditioning (PC) in a pH-sensing-relevant range. They employed various techniques, including voltammetry, scanning electron microscopy (SEM), energy-dispersive X-ray spectroscopy (EDS), X-ray diffraction (XRD), and X-ray photoelectron spectroscopy (XPS). This article provides valuable insights. Thus, I can recommend it for publication in Sensors. And I have listed a few minor suggestions below for further improvement:

1.      The authors mention that “the focus of this work is to understand and interpret the correlation between material state and electrode conditioning potentials within pH sensing regimes.” I would like to understand the mechanisms by which between the material state and electrode conditioning potentials. Please elaborate further or provide additional reference.

2.      The authors mention that “the performance of electrodeposited IrOx pH sensors is still inconsistent for rigorous research use and commercialization.” Have the authors compared this research to commercial sensors? Is the study theoretically consistent with commercial sensors? Please elaborate on the advantages of this research in future developments.

3.      Please enhance the clarity of the images, particularly figures 1, 5, and 6.

4.      The authors mention that “This suggests that “safe” potentials (i.e. minimally altering of surface bonding) are between 0.0 V and +0.4 V.” I would like to know if there is potential for integrating the currently known parameters into various application systems. Please elaborate further on this.

5.      The authors mention that “Despite continued uncertainty in exact speciation, the results here are valuable.” There are numerous aspects of the research that even the authors are uncertain about. Regarding the reproducibility of experimental data, whether it is due to insufficient sample size, and if there are better approaches to achieve 'certainty' in the results.

Author Response

Dear reviewer,

The attachment is our response. Thank you very much for your time and valuable comments.

Reviewer 2 Report

Comments and Suggestions for Authors

In the manuscript entitled as "Polarization Conforms Performance Variability in Amorphous Electrodeposited Iridium Oxide pH Sensors: A Thorough Surface Chemistry Investigation" investigated the influence of polarization on the performance variability of amorphous electrodeposited iridium oxide pH sensors. Through surface chemistry investigation, the study revealed interactions between the surfaces. The authors found that polarization process can control the consistency of sensor performance by altering surface chemical bonding. Although the exact species still remain uncertain, this research is of significant importance for understanding and explaining the relationship between material state and electrode conditions during polarization process, and provides baseline data for future studies on ion deposition.

This article represents a valuable study which provides an in-depth understanding of the impact of polarization on the performance variability in amorphous electrodeposited iridium oxide pH sensors. The research is comprehensive and involves multiple surface analysis techniques, including XPS, XRD, SEM, and EDS. These techniques complement each other and provide a comprehensive understanding of surface chemical changes. The impact of polarization on performance consistency is discussed in detail. For example, as the polarization potential increases, the oxidation state of the surface increases, and the degree of hybridization between iridium and oxygen also increases. In any case, despite its qualities, the paper has shortcomings which must be addressed before the manuscript can be published.

1.         Some of the image qualities are relatively low, such as the attenuation curves and EDS patterns, which may impact the interpretation and analysis of the data. It is recommended that the authors provide higher quality images to better showcase the experimental results.

2.         The detailed preparation and experimental methods in section 2. Materials and Methods are discussed in the supplementary materials. This may require readers to spend additional time and effort consulting the supplementary materials in order to understand the experimental process and results.

3.         Although this study provides valuable information about the surface chemistry and polarization processes of the material, there are still limitations in the quantitative analysis of the data. For example, the variation trend of the Ir:O ratio under different polarization conditions may differ from other studies in the literature, which could affect the understanding of the material's surface chemistry. Additionally, there are multiple possible explanations for the work function spectra, leading to uncertainties in the understanding of the mechanisms. Therefore, despite the progress made in surface chemistry in this study, further research and validation are still needed for precise species and mechanisms.

4.         The conclusion section of the paper mainly focuses on the theoretical significance and practical application value of the experimental results, but provides limited suggestions and discussions on how to further optimize the experimental methods and improve the experimental efficiency.

In conclusion, the paper is recommended to be published after minor revision.

Author Response

(The authors gave the same response as above.)
